# Entropy, Entanglement Islands and Swampland Bounds in the Karch-Randall Braneworld

**Hao Geng**

*Center for the Fundamental Laws of Nature, Harvard University, 17 Oxford St., Cambridge, MA, 02138, USA.*

*E-mail:* haogeng@fas.harvard.edu

ABSTRACT: Motivated by the study of entanglement island in the Karch-Randall braneworld, it has been conjectured and proven in general that entanglement island is not consistent with long-range (massless) gravity. In this paper, we provide a careful check of this conclusion in a model of massless gravity that is constructed using the Karch-Randall braneworld. We show that there is indeed no entanglement island and hence not a nontrivial Page curve due to the diffeomorphism invariance if we are studying the correct question which is though subtle. Moreover, we show that this conclusion is not affected by deforming the setup with the Dvali–Gabadadze–Porrati (DGP) terms. Furthermore, we show that the consistency of holography in this model will provide nontrivial constraints to the DGP parameters. This study provides an example that causality and holography in anti-de Sitter space can be used to constrain low energy effective theories. At the end, we clarify several subtleties in the braneworld gravity which are overlooked in the literature and we discuss the robustness of the above results against possible coarse-graining protocols to define a subregion in a gravitational theory.

## 1  Introduction

The Karch-Randall braneworld [1, 2] considers a very simple scenario— embedding a brane whose geometry is asymptotically $\text{AdS}_d$ in a bulk spacetime with asymptotically $\text{AdS}_{d+1}$ geometry. It however provides the playground to study long-standing questions in quantum gravity such as the black hole information paradox and it has taught us many important lessons about quantum gravity. This is all due to its doubly holographic nature [3–5].

One example of the aforementioned progress is the calculation of the Page curve of the black hole radiation. The original version of the Karch-Randall braneworld provides the holographic dual of the set-up that is used to define and calculate the entanglement entropy of the black hole radiation (see Fig.1). This set-up considers coupling the asymptotically (large) $\text{AdS}_d$ black hole spacetime to a thermal bath which is modeled by a d-dimensional conformal field ($\text{CFT}_d$) theory on a half Minkowski space. The coupling is achieved by gluing the asymptotic boundary of the black hole spacetime along the boundary of the half-space $\text{CFT}_d$ by imposing transparent boundary condition for the energy flux. The black hole radiation can be modeled in this set-up as a subregion $\mathcal{R}$ of the bath $\text{CFT}_d$. Then we can calculate the time-dependence of this subregion entanglement entropy as the entanglement entropy of the black hole radiation (see Fig.1). In the Karch-Randall braneworld, the $\text{AdS}_d$ black hole can be thought of as the brane and the $\text{AdS}_{d+1}$ bulk provides the holographic dual of

this d-dimensional set-up. The aforementioned entanglement entropy can be calculated in the $\mathrm{AdS}_{d+1}$ bulk using the Ryu-Takayanagi surface [6, 7]. The calculation shows that at late times the entanglement wedge of $\mathcal{R}$ contains a region $\mathcal{I}$ on the brane that is disconnected from $\mathcal{R}$ and this region is defined as the entanglement island of $\mathcal{R}$ and moreover the entanglement entropy of $\mathcal{R}$ follows a unitary Page curve [4, 5]. In this calculation, the late time emergence of the entanglement island is essential to have the unitary Page curve [3, 8, 9]. Nevertheless, the original version of the Karch-Randall braneworld also captures another important property of the d-dimensional set-up we mentioned that is that the graviton in the $\mathrm{AdS}_d$ black hole spacetime becomes massive due to the bath coupling [10]. This can be seen in the Karch-Randall braneworld by studying the Klauza-Klein spectrum of the $\mathrm{AdS}_{d+1}$ graviton [1]. Motivated by this observation it was conjectured and proved that entanglement island cannot exist in massless gravity theories for a large class of situations including asymptotically AdS spacetimes [11–13].

Interestingly, before the general proof was given in [13], it was shown in [12] that entanglement island doesn't exist in a massless gravity theory that is constructed as a modification of the original version of the Karch-Randall braneworld. This was called *wedge holography* in [14]. In this scenario we embed two Karch-Randall branes $\mathcal{M}_d^{(L)}$ and $\mathcal{M}_d^{(R)}$ (i.e. with asymptotically $\mathrm{AdS}_d$ geometries) into an asymptotically $\mathrm{AdS}_{d+1}$ bulk such that the two branes form the boundary of a wedge $\mathcal{W}_{d+1}$ (see Fig.2). This system is described by the following action

$$S_1 = -\frac{1}{16\pi G_{d+1}} \int_{\mathcal{W}_{d+1}} d^{d+1}x\sqrt{-g}(R - 2\Lambda) - \frac{1}{8\pi G_{d+1}} \int_{\mathcal{M}_d^{(L)} \cup M_d^{(R)}} d^d x\sqrt{-h}(K - T)\,, \quad (1.1)$$

where $K$ is the trace of the extrinsic curvature on the brane (which in general takes different values on the two branes), $h_{\mu\nu}$ is the induced meric on the brane, $T$ is the tension of the brane (which in general takes different values on the two branes) and $\Lambda = -\frac{d(d-1)}{2L^2}$ (later we will set the AdS length scale $L = 1$ for convenience) is the bulk cosmological constant. This system has three equivalent descriptions:

1. **Bulk description**: Einstein gravity in an $\mathrm{AdS}_{d+1}$ space $\mathcal{M}'_{d+1}$ containing two $\mathrm{AdS}_d$ branes $\mathcal{M}_d^{(L)}$ and $\mathcal{M}_d^{(R)}$,which intersect each other on the asymptotic boundary (we call the place they intersect as the defect and it is (d-1)-dimensional) and therefore form the boundary of a wedge $\mathcal{W}_{d+1}$ (see Fig.2);

2. **Intermediate description**: Two $d$-dimensional CFTs coupled to gravity on distinct asymptotically $\mathrm{AdS}_d$ spaces $\mathcal{M}_d^L$ and $\mathcal{M}_d^R$, with these systems being connected via a transparent boundary condition at a defect $\mathcal{M}_{d-1}^{(0)} = \partial\mathcal{M}_d^L = \partial\mathcal{M}_d^R$;

3. **Boundary description**: A $(d-1)$-dimensional CFT on $\mathcal{M}_{d-1}^{(0)}$.

A consistent background of such system has to satisfies three sets of differential equations– the bulk Einstein's equation and the two brane embedding equations with Neumann boundary condition for bulk metric fluctuations near the brane. It was shown in [12] that there is no

entanglement island in one such consistent background that is relevant to the study of black hole information paradox (as will be reviewed in Sec.2). The essential reason was that now the subregion $\mathcal{R}$ is on the brane $\mathcal{M}_d^{(R)}$ and due to diffeomorphism invariance of massless gravity the subregion $\mathcal{R}$ itself should be determined dynamically. It was shown in [12] that a natural dynamical principle gives a vanishing $\mathcal{R}$ as well as its entanglement island $\mathcal{I}$.

Nevertheless, [15, 16] claimed that this conclusion that there is no entanglement island in this wedge holography set-up can change by adding appropriately chosen DGP terms

$$S_{DGP} = -\frac{1}{16\pi G_d^{(L)}} \int_{\mathcal{M}_d^{(L)}} d^d x \sqrt{-h} R_B - \frac{1}{16\pi G_d^{(R)}} \int_{\mathcal{M}_d^{(R)}} d^d x \sqrt{-h} R_B \,, \qquad (1.2)$$

where $R_B$ is the d-dimensional Ricci scalar on the brane. This modification adds internal gravitational dynamics on the branes such that in the intermediate picture the theory is no longer purely induced from the bulk.

The first aim of this paper is to carefully clarify the relevant question to entanglement island we asked in [12] in each description of the wedge holography set-up. This can be used to show that there is indeed no entanglement island with physically reasonable values of $G_d^{(L)}$ and $G_d^{(R)}$. Moreover, this motivates another basic consideration that can be used to further constrain the values of $G_d^{(L)}$ and $G_d^{(R)}$. This consideration is based on the consistency of wedge holography in the boundary description. As we will see that this result should be understood as the statement that causality and holography in anti-de Sitter space can be used to constrain low energy effective theories. This is the extension of the earlier program of using holography to constrain low energy effective theories in de Sitter space [17] to anti-de Sitter space.[1] The second aim of this paper is to give a precise description of the intermediate picture and discuss possible coarse-graining protocols under which we may be able to define a subregion in a gravitational theory and hence may have entanglement island and the problems of these protocols.

The paper is organized as follows. In Sec.2 we will review our study in [12] to carefully clarify the questions we considered in each description of the wedge holography. We will use this to show that there is in fact no entanglement island in the set-up considered in [15, 16] due to diffeomorphism invariance. Then we show that the consideration in the boundary description will give us two criteria to further constrain the parameters in $G_d^{(L)}$ and $G_d^{(R)}$. In Sec.4 we will derive the precise description of the intermediate description and show how we could derive island formula in this description. Moreover, we discuss two classes of coarse-graining protocols that is usually claimed to be able to define a subregion in a gravitational universe and relax the diffeomorphism invariance constraint on the existence of entanglement island in the literature [23, 24]. We show that it's subtle for them to be consistently defined and once they are consistently defined we either lose entanglement island or have a nontrivial modification of the gravitational theory. At the end, we will conclude in Sec.5.

---

[1]See [18–22] for other examples of using holography to constrain low energy effective theories in anti-de Sitter space.

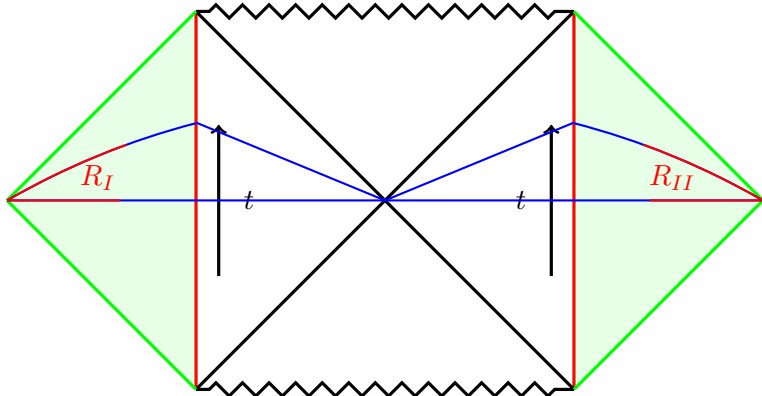

**Figure 1**: We show the Penrose diagram of an eternal black in $\mathrm{AdS}_d$ coupled to $d$-dimensional baths that is used in the calculation of the Page curve of black hole radiation. The baths are the shaded green region and the geometry of the baths are $d$-dimensional Minkowski space. We specify the two red vertical lines as the conformal boundary of the $\mathrm{AdS}_d$ black hole. We choose time evolution as indicated in the diagram. We also specify two Cauchy slices of this time evolution as the two blue curves and on each of them we denote the subsystem $R = R_I \cup R_{II}$ in red. We emphasize that the Cauchy slices of this time evolution all go through the bifurcation horizon so they don't touch the black interior.

## 2    No Entanglement Island in Massless Gravity

In this section, we review the result that there is no entanglement island in the wedge holography model of massless gravity in [12]. We will clarify the question that we considered in the intermediate description and show that introducing DGP terms doesn't in fact change this conclusion. At the end, we will consider the question in the boundary description and see that it could be used to constrain DGP terms in wedge holography.

### 2.1    The Set-up

As we discussed in the introduction, a consistent background where we could study wedge holography should satisfy

$$\delta S_1 = 0 \,, \tag{2.1}$$

where $S_1$ is given by Equ. (1.1) and the boundary condition for the metric fluctuation is of the Neumann type:

$$\delta g_{\mu\nu} \neq 0 \,, \quad i = L, R \tag{2.2}$$

where $n_i$ denotes the unit normal direction of the (two) brane(s). This gives us three equations

$$R_{\mu\nu} - \frac{1}{2} g_{\mu\nu} R + \Lambda g_{\mu\nu} = 0 \,, \quad K_{\mu\nu} = (K - T) h_{\mu\nu} \,, \tag{2.3}$$

where the second equation should be satisfied for each brane with their own values of tension and $h_{\mu\nu}$ is the induced metric on the brane.

A general solution of all these three equations is hard to find either due to the complicated backreaction of the brane to the bulk geometry [4] or the nonstationarity of the branes in a generic bulk solution of the Einstein's equation (the first equation) [25]. Nevertheless, it is known that a nice natural solution exists which can be used to study black hole physics in wedge holography [12]. It is known as a *black string*. The bulk metric is given by

$$ds^2 = \frac{1}{u^2 \sin^2 \mu} \left[ -h(u)dt^2 + \frac{du^2}{h(u)} + d\vec{x}^2 + u^2 \, d\mu^2 \right], \quad h(u) = 1 - \frac{u^{d-1}}{u_h^{d-1}}, \tag{2.4}$$

where the polar coordinate $\mu$ takes its value in $(0, \pi)$, $\{\mu = 0\} \cup \{\mu = \pi\}$ corresponds to the asymptotic boundary and $u_h$ is the parameter parametrizing the temperature of the black string. The two branes are located at constant-$\mu$ hypersurfaces $\mu = \mu_L < \frac{\pi}{2}$ and $\mu = \mu_R > \frac{\pi}{2}$ and their tensions are given by

$$T_L = (d-1)\cos \mu_L, \quad T_R = -(d-1)\cos \mu_R, \tag{2.5}$$

which are both positive.

We can see that this (d+1)-dimensional geometry is a foliation of d-dimensional AdS Schwartzschild black holes with a foliation parameter $\mu$. The two branes will intersect each other at $u = 0$ (a (d-1)-dimensional manifold) on the asymptotic boundary. Since the two branes are of positive tension, they will cutoff the bulk region from them to their closest part of the asymptotic boundary ($\mu = 0$ for left brane and $\mu = \pi$ for right brane). Hence the left over region in the bulk is a (d+1)-dimensional wedge bounded by the two branes and the geometries on the two branes are $\text{AdS}_d$ Schwartzschild black holes. (Fig.2.) We notice that the intermediate picture Penrose diagram of this case is different from the usual case with a flat bath Fig.1.

## 2.2  The Question and the Result without DGP Terms

The question we asked in [12] is that in the intermediate picture whether we could find entanglement island $\mathcal{I}$ on the left brane for a well-defined subregion $\mathcal{R}$ on the right brane. To do so we considered $\mathcal{R}$ to be of the type that is relevant to the computation of Page curve in the Karch-Randall braneworld. That is that $\mathcal{R}$ should straddle the two exteriors of the black hole geometry on the right brane (see Fig.3). Moreover, for $\mathcal{R}$ itself to be well-defined and consistent with diffeomorphism invariance in massless gravity we defined it dynamically in [12] by minimizing the entropy functional over both $\mathcal{I}$ and $\mathcal{R}$:

$$S = \min_{\mathcal{I}, \mathcal{R}} \text{ext} \, S_{\text{gen}}(\mathcal{R} \cup \mathcal{I}), \quad S_{\text{gen}}(\mathcal{R} \cup \mathcal{I}) = S_{\text{matter}}(\mathcal{R} \cup \mathcal{I}). \tag{2.6}$$

Here we notice that the generalized entropy doesn't contain the usual area term due to the fact that there is no internal gravitational dynamics on the branes and, as we will discuss later (see Sec.4.1 for details), this would be changed with DGP terms added. A more careful definition of $\mathcal{R}$ and how Equ. (2.6) can be derived is discussed in Sec.4.2. The power of the Karch-Randall braneworld models is that we can use the (d+1)-dimensional bulk to compute

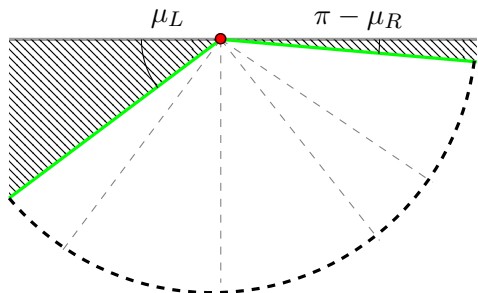

**Figure 2**: The black string in wedge holography. We embed two KR branes in the setup as the two solid green lines. The shaded region behind them is cutoff. The physical space in the bulk is the wedge-shaped region bounded by the two branes. The two branes intersect at the red defect on the asymptotic boundary of the bulk. The dashed black line is the black string horizon separating the exterior and interior regions. The dashed green lines are hypersurfaces of constant $\mu$. The geometry on each constant-$\mu$ slice, including the two branes, is the AdS$_d$ Schwartzschild black hole. We show one asymptotic exterior of the geometry and we always put in mind that the geometry we consider is actually maximally extended which has two exteriors.

the highly nontrivial quantity $S_{\text{matter}}(\mathcal{R} \cup \mathcal{I})$. It can be computed by the Ryu-Takayanagi formula [6] by looking for entangling surfaces that are homologous to $\mathcal{R} \cup \mathcal{I}$ and using the one $\gamma$ that minimizes the area functional to be the one that compute $S_{\text{matter}}(\mathcal{R} \cup \mathcal{I})$:

$$S = \min_{\mathcal{I},\mathcal{R}} \text{ext} \frac{A(\gamma)}{4G_{d+1}} \,, \tag{2.7}$$

where $A(\gamma)$ is the area of the minimal surface $\gamma$. This formula tells us that we have to further minimize over the possible ending points for the area functional of minimal area surface $\gamma$ and the output of this minimization is the entanglement island $\mathcal{I}$ and a well-defined subregion $\mathcal{R}$.

As it is standard for disconnected boundary subregions in AdS/CFT correspondence, there two possible topologies of the minimal area surface $\gamma$. One gives a connected entanglement wedge of $\mathcal{R} \cup \mathcal{I}$ in the bulk and the other gives a disconnected one (see Fig.4). We call them $\gamma_c$ and $\gamma_{dc}$ respectively. $\gamma_c$ has two components one in each exterior of the bulk wedge and $\gamma_{dc}$ has two components one close to each brane and all go through the interior of the black string (if $\mathcal{R} \cup \mathcal{I}$ is non-empty). Moreover, $\gamma_c$ is confined to a constant time slice in $t$ of Equ. (2.4) and $\gamma_{dc}$ is not confined to such a slice and its area depends on $t$ where $t$ is the time-coordinate of the Cauchy slice (constant-$t$ slice) we take to define the subregion $\mathcal{R}$ as shown in the intermediate picture Penrose diagram Fig.3 (the corresponding diagram in the bulk is in Fig.4). We proved in [12] that with the minimization procedure in Equ. (2.7) done each component of $\gamma_c$ must be the black string horizon and so the entanglement island $\mathcal{I}$ and $\mathcal{R}$ must both be empty. Hence there can be no entanglement island in this model.

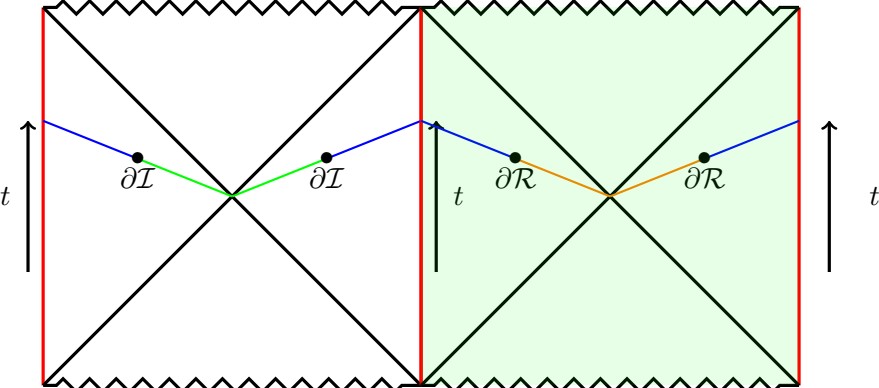

**Figure 3**: The Penrose diagram of the intermediate picture. We have two black holes (i.e the geometry on the two branes) coupled to each other by gluing them along their common asymptotic boundaries. The left asymptotic boundary of the left black hole is glued with the right asymptotic boundary of the right black hole. We take a constant time slice (the blue slices union the green and orange slices) of the time evolution that we are considering in the calculation of entanglement island and Page curve. The time is defined on the asymptotic boundary (the defects). The orange region is a putative subregion $\mathcal{R}$ and the green region is a putative entanglement island $\mathcal{I}$ of the subregion $\mathcal{R}$. As opposed to the situation in Fig.1 the bath (the green shaded region) is now an $\text{AdS}_d$ black hole.

Nevertheless, we should be careful about the value we get from Equ. (2.7) as it computes the fine-grained entanglement entropy of the subregion $\mathcal{R}$. If $\mathcal{R}$ is empty then $S$ should be zero but this is not consistent with the result we would obtain if we use $\gamma_c$ as the union of two horizons. To resolve this issue, we should also consider $\gamma_{dc}$. The output of the minimization Equ. (2.7) for $\gamma_{dc}$ is that $\gamma_{dc}$ shrinks all the way to the union of two points and is hence empty which gives 0— the smallest possible value of $S$ according to Equ. (2.7). Therefore, we still have the result that both $\mathcal{I}$ and $\mathcal{R}$ are empty and we also have the physically consistent result that the resulting entanglement entropy calculated by Equ. (2.7) is zero.

## 2.3 Adding DGP Terms Doesn't Change the Result

If we have DGP terms Equ. (1.2) added to the system Equ. (1.1), the entropy functional $S_{\text{gen}}(\mathcal{R} \cup \mathcal{I})$ would have additional area terms than just $S_{\text{matter}}(\mathcal{R} \cup \mathcal{I})$.[2] As a result, instead of Equ. (2.6) we would have

$$S = \min_{\mathcal{I},\mathcal{R}} \text{ext} \, S_{\text{gen}}(\mathcal{R} \cup \mathcal{I}), \quad S_{\text{gen}}(\mathcal{R} \cup \mathcal{I}) = S_{\text{matter}}(\mathcal{R} \cup \mathcal{I}) + \frac{A(\partial \mathcal{I})}{4G_d^{(L)}} + \frac{A(\partial \mathcal{R})}{4G_d^{(R)}}, \qquad (2.8)$$

---

[2]Moreover, the brane embedding equation will be different from the second equation in Equ. (2.3) with a d-dimensional Einstein term added. However, it can be shown that the constant$-\mu$ slices of Equ. (2.4) still satisfy the resulting brane embedding equation with the tensions modified comparing to Equ. (2.5) [26–28].

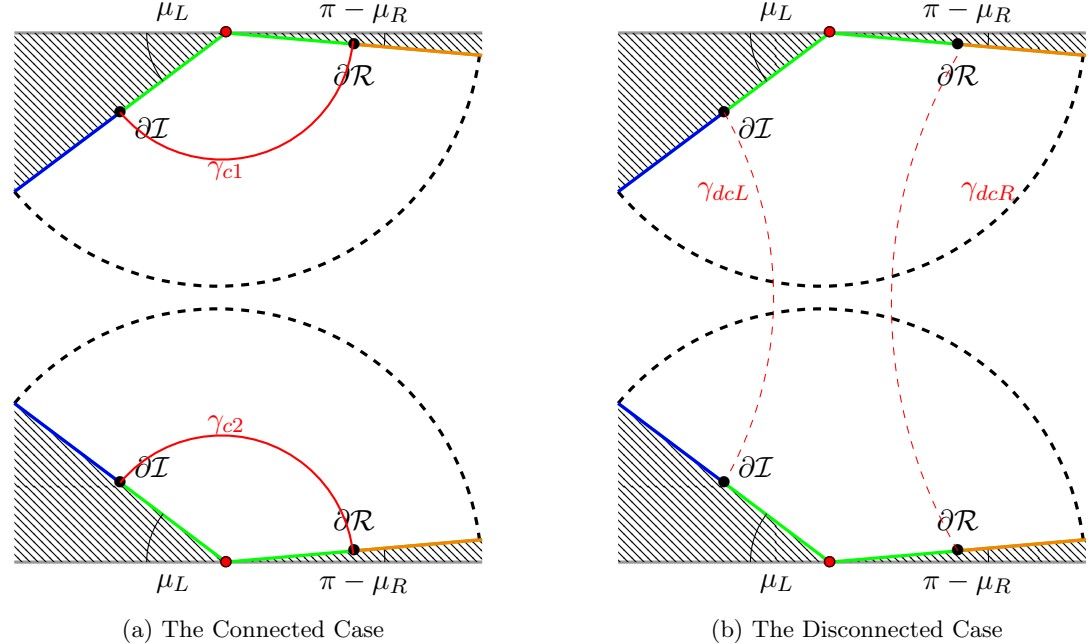

(a) The Connected Case           (b) The Disconnected Case

**Figure 4**: The bulk diagram indicating the bulk configuration corresponding to the constant time slice in Fig.3. The two horizons in each panel should be identified as the horizon on each panel should be the bifurcation horizon. The entangling surface $\gamma_c = \gamma_{c1} \cup \gamma_{c2}$ in the connected case lives on this constant time slice (and we use solid red lines for them). Nevertheless, the entangling surface $\gamma_{dc} = \gamma_{dcL} \cup \gamma_{dcR}$ is not confined on this constant time slice and is time-dependent (we draw them by dashed red lines to indicate the fact that they are not confined on this diagram).

where $A(\partial\mathcal{I})$ and $A(\partial\mathcal{R})$ are the area of the boundary of the entanglement island $\mathcal{I}$ and the subregion $\mathcal{R}$. Similar to Equ. (2.7) this quantity can be computed purely geometrically where we could use the Ryu-Takayanagi formula to replace $S_{\text{matter}}(\mathcal{R} \cup \mathcal{I})$ by $\frac{A(\gamma)}{4G_{d+1}}$:

$$S = \min_{\mathcal{I},\mathcal{R}} \text{ext} \left( \frac{A(\gamma)}{4G_{d+1}} + \frac{A(\partial\mathcal{I})}{4G_d^{(L)}} + \frac{A(\partial\mathcal{R})}{4G_d^{(R)}} \right), \qquad (2.9)$$

where $\gamma$ is a bulk minimal area surface that is homologous to $\mathcal{R} \cup \mathcal{I}$ on the two branes. Again there are two possible topologies of $\gamma$— $\gamma_c$ connects $\partial\mathcal{I}$ and the left brane to $\partial\mathcal{R}$ on the right brane and $\gamma_{dc}$ has two disconnected components homologous to $\mathcal{I}$ and $\mathcal{R}$ separately.

Nevertheless, for the interpretation of Equ. (2.9) as calculating entanglement entropy we have to ensure that the functional to be minimized is positive for any minimal area surface $\gamma$, otherwise the result of the minimization would give us a negative value which cannot be interpreted as the entanglement entropy of a quantum state. For such choices of the parameters $G_{d+1}$, $G_d^{(L)}$ and $G_d^{(R)}$ the result of the minimization is again that both $\mathcal{R}$ and $\mathcal{I}$

are empty (the orange and blue region in Fig.3 all shrink to zero) and $S = 0$. This tells us that there is no entanglement island.

## 2.4 A Question in the Boundary Description

In Sec.2.2 and Sec.2.3, we studied a question that is relevant to entanglement island in the intermediate picture of the wedge holography and we got consistent results from holography with the general principle that entanglement island is inconsistent with long-range gravity [13]. However, as we reviewed in the introduction, the wedge holography has three equivalent descriptions and for the internal consistency of wedge holography we also have to make sure that the boundary description is self-consistent.

In the boundary description, the black string Equ. (2.4) with the the branes Equ. (2.5) is described by the thermal-field double (TFD) state of two $\mathrm{CFT}_{d-1}$'s each lives on one asymptotic defect [12]. Moreover, the entanglement entropy $S_{\mathrm{defect}}$ between the two defects in this state is the thermal entropy for each of the defects and it is captured by the black string horizon $\mathcal{H}$ and is computed as

$$S_{\mathrm{defect}} = \frac{A_{\mathcal{H}}}{4G_{d+1}} + \frac{A_{\partial\mathcal{H}_L}}{4G_d^{(L)}} + \frac{A_{\partial\mathcal{H}_R}}{4G_d^{(R)}} , \qquad (2.10)$$

where $\partial\mathcal{H}_L$ and $\partial\mathcal{H}_R$ denotes the induced horizons on the left and right branes respectively. In the absence of DGP terms the expression only contains the first term. A basic consistency of wedge holography is that the result Equ. (2.10) should be consistent with the Ryu-Takayanagi formula. The Ryu-Takayanagi formula tells us that $S_{\mathrm{defect}}$ is computed by

$$S = \min_{\partial\gamma_R, \partial\gamma_L} \mathrm{ext} \left( \frac{A(\gamma)}{4G_{d+1}} + \frac{A(\partial\gamma_R)}{4G_d^{(L)}} + \frac{A(\partial\gamma_L)}{4G_d^{(R)}} \right) , \qquad (2.11)$$

where $\gamma$ is a bulk minimal area surface that connects the two branes and homologous to the defect and $\partial\gamma_R$ and $\partial\gamma_L$ are the cross-section of $\gamma$ with the right and left branes respectively (see Fig.5). In other words, the consistency of wedge holography would require that the output of the minimization in Equ. (2.11) is that the entangling surface $\gamma$ should be the black string horizon $\mathcal{H}$.

For setups (combinations parameters $G_{d+1}$, $G_d^{(L)}$ and $G_d^{(R)}$) where the the output of the minimization in Equ. (2.11) is not the black string horizon $\mathcal{H}$, the wedge holography cannot be consistently defined or these setups are not consistent with wedge holography. Moreover, for these setups the entanglement wedge of the defect determined by the RT surface $\gamma_{RT}$ would be smaller that the black string exterior that include the defect we are considering. This is the same as saying that the causal wedge of the defect is bigger than its entanglement wedge. Hence the causality and holography are not consistent in these setups and these setups should be in the swampland.

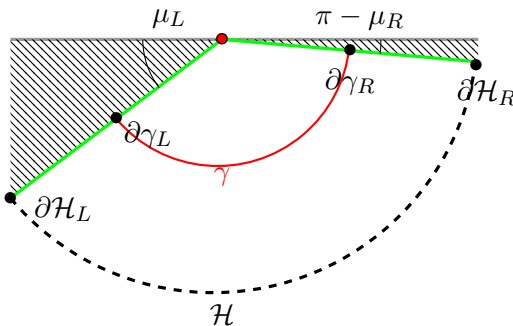

**Figure 5**: The calculation of the entanglement entropy of one of the defects. The red surface $\gamma$ is a putative output of the minimization in Equ. (2.11). For consistency of the wedge holography $\gamma$ should in fact be the black string horizon i.e. the black dashed line.

## 3 Consistency of Wedge Holography and the Swampland Bounds

From our study in Sec.2.4 we see that the consistency of wedge holography requires that in the background geometry Equ. (2.4) the black string horizon $\mathcal{H} : u = u_h$ should compute the entanglement entropy between the two defects in the TFD state. The formula of this entanglement entropy is given by Equ. (2.10). Moreover, this result should be consistent with the Ryu-Takayanagi formula which tells us that the black string horizon $\mathcal{H}$ should be the solution of the minimization problem defined in Equ. (2.11).

These consistency conditions convey nontrivial messages as they input important constrains on the DGP parameters $G_d^{(L)}$ and $G_d^{(R)}$. Firstly, the entanglement entropy between the two defects has a clear statistical meaning as the Hilbert spaces of the two defects are cleanly factorized as a tensor product of two $\text{CFT}_{d\text{-}1}$ Hilbert spaces.[3] This tells us that the entanglement entropy between them should be positive and finite (upto an IR divergent factor due to the infinity volume). In other words the algebras associated with the two defects are Type I Von-Neumann algebras [31–37] which again tells us that the entanglement entropy between them is positive and finite. Moreover, the requirement that the answer of the minimization problem Equ. (2.11) should be the black string horizon $\mathcal{H}$ gives another constraint. This requirement can be understood as a direct consequence of the entanglement wedge reconstruction of holography which in our case says that the physics of the defect is fully captured by the physics in the bulk region $\Sigma_{\gamma_{\min}}$ inclosed by the answer $\gamma_{\min}$ of the minimization problem Equ. (2.11) (the defect is on the boundary of $\Sigma_{\gamma_{\min}}$). This says that the bulk domain of dependence of $\Sigma_{\gamma_{\min}}$ should include or equal to the bulk domain of dependence of the defect and meanwhile doesn't intersect the bulk domain of dependence of the other defect. In other

---

[3]Here is a caveat that we are literally considering the Hilbert space of a $\text{CFT}_{d-1}$ which means that we shouldn't take the infinite central charge $c \to \infty$ limit. The reason is that in this limit the Hilbert space is generated as a Fock space by single-trace operators and the resulting theory doesn't behave as a $\text{CFT}_{d-1}$ due to the Hagedorn growth of the partition function at high temperature [29, 30].

words, this requirement comes from the consistency between holography and causality. This condition exclusively determines that $\gamma_{\min} = \mathcal{H}$.

In summary, we have two swampland constraints on the DGP parameters $G_d^{(L)}$ and $G_d^{(R)}$ from the consistency of wedge holography

- **Condition 1:** $S_{\text{defect}} = \frac{A_{\mathcal{H}}}{4G_{d+1}} + \frac{A_{\partial \mathcal{H}_L}}{4G_d^{(L)}} + \frac{A_{\partial \mathcal{H}_R}}{4G_d^{(R)}} \geq 0$.

- **Condition 2:** The answer of the minimization problem on the RHS of Equ. (2.11) is the black string horizon i.e. $\gamma_{\min} = \mathcal{H}$.

Here we quickly exploit the first constraint. In the background geometry Equ. (2.4), we can compute

$$
\begin{aligned}
&\frac{A_{\mathcal{H}}}{4G_{d+1}} + \frac{A_{\partial \mathcal{H}_L}}{4G_d^{(L)}} + \frac{A_{\partial \mathcal{H}_R}}{4G_d^{(R)}} \\
=&V_{d-2}\left( \frac{1}{4G_{d+1}} \int_{\mu_L}^{\mu_R} \frac{d\mu}{u_h^{d-2} \sin^d \mu} + \frac{1}{4G_d^{(L)}} \frac{1}{u_h^{d-2} \sin^{d-1} \mu_L} + \frac{1}{4G_d^{(R)}} \frac{1}{u_h^{d-2} \sin^{d-1} \mu_R} \right),
\end{aligned}
\tag{3.1}
$$

where $V_{d-2}$ is the transverse space volume and is always positive. Hence we have the following constraint

$$
\frac{1}{4G_{d+1}} \int_{\mu_L}^{\mu_R} \frac{d\mu}{u_h^{d-2} \sin^d \mu} + \frac{1}{4G_d^{(L)}} \frac{1}{u_h^{d-2} \sin^{d-1} \mu_L} + \frac{1}{4G_d^{(R)}} \frac{1}{u_h^{d-2} \sin^{d-1} \mu_R} \geq 0.
\tag{3.2}
$$

This constraint can be exploited numerically for a given dimension $d$. We defer such an detailed exploration together with that of the **Condition 2** to [28].

## 4 The Intermediate Picture and Coarse-Graining

In this section, we study more details of the intermediate picture. We firstly show the precise description of the intermediate picture. Then we show how Equ. (2.6) as well as Equ. (2.8) can be derived where the minimization over $\mathcal{R}$ is a direct consequence of diffeomorphism invariance. At the end, we discuss possible coarse-graining protocols that we can take $\mathcal{R}$ in Equ. (2.6) and Equ. (2.8) as a fixed region (i.e. we don't have to minimize over $\mathcal{R}$) and the problem of these protocols.

### 4.1 The Precise Description of the Intermediate Picture

For the sake of convenience, let's firstly consider the case without the DGP terms which is the conventional description of the braneworld models [1, 38, 39]. Motivated by the early studies of holography and braneworld [40], it is usually stated that the intermediate picture of the Karch-Randall braneworld is described as a system of a conformal field theory with a UV-cutoff and coupled to dynamical gravity in $\text{AdS}_d$ meanwhile this system in $\text{AdS}_d$ is coupled to a d-dimensional bath glued along the conformal boundary of the $\text{AdS}_d$. The bath can be

gravitational or nongravitational, for example in the wedge holography model Equ. (1.1) the bath is gravitational and in the original version of Karch-Randall braneworld [1] the bath is nongravitational.

This conclusion comes from the observations that first the Karch-Randall brane cuts off part of the bulk geometry and second the boundary condition for bulk metric fluctuation close to the brane is of the Neumann type Equ. (2.2). Combined with the AdS/CFT correspondence [41–43] the first observation tells us that the matter field on the brane is dual to the bulk gravity and hence is a CFT with a UV cutoff and the cutoff scale is set by the bulk brane position $\mu_B$. The second obervation tells us that the metric on the brane is fluctuating i.e. the intermediate picture is a gravitational theory.

Nonetheless, it is never made clear what this conclusion precisely means for example given the $\text{CFT}_d$ that is dual to the bulk $\text{AdS}_{d+1}$ gravity (without the brane) then how one could get the theory on the brane (i.e. the intermediate picture) when the brane is introduced.

The statement that the $\text{CFT}_d$ would then have a UV cutoff presumably means that the matter theory in the intermediate picture is from an irrelevant deformation of that $\text{CFT}_d$ and the cutoff scale denotes the strength of this deformation. Moreover, such a deformation should be different from the usual picture of the renormalization group (RG) flow where the UV cutoff scale comes from momentum space Wilsonian procedure. There are two reasons that they are different: 1) The Wilsonian procedure hides the UV details of the theory into the effective coupling constants and the physics below the UV cutoff scale is insensitive to this operation. This would predict that a light signal send from the bulk to the brane will be reflected back to the bulk just like the brane is not there.[4] This clearly doesn't make sense as this signal absorbing and emission procedure is highly nonlinear in the intermediate picture and there is no reason that its bulk dual would be as simple as the above expectation. 2) Fine-grained quantities such as the replica entropy[5] is not sensitive to the Wilsonian procedure [45]. This suggests that the subregion replica entropy is invariant under the RG flow. This clearly contradicts the calculation based the RT formula in the braneworld holography where the RT surface computing the replica entropy[6] never penetrates the brane but instead ending on it which gives a cutoff scale dependent answer for the replica entropy. This issue is partly resolved by the recent study of the holographic dual of the $T\bar{T}$-deformation of CFT [47–51] where the CFT is deformed in a controllable way and the entanglement entropy is shown to be non-invariant under the deformation [52]. Hence, the matter field in the intermediate

---

[4]More precisely, before the Wilsonian procedure there is a deformation of the $\text{CFT}_d$ introduced. Such deformations in AdS/CFT are generically multitrace deformations and their holographic duals are known [44] as modifying the boundary conditions of bulk fields.

[5]The replica entropy is the entropy calculated by the replica trick which however takes into account the conical singularity of the replica manifold. For quantum field theory its value is different from entanglement entropy if the theory is non-minimally coupled with the metric. The replica entropy is called the black hole entropy in [45]. The replica entropy is Wilsonian RG invariant but the entanglement entropy is not [45].

[6]Analogous to the AdS/CFT results [46], using the replica trick and the holographic duality the replica entropy in the intermediate picture is mapped to the area of the bulk RT surface with ending points on the brane.

picture is the holographic dual of the cutoff $\text{AdS}_{d+1}$ bulk and it is described by a $T\bar{T}$-like deformation of the $\text{CFT}_d$ on the brane.

Furthermore, the statement that the brane part of the intermediate picture is gravitational means that in the path integral description of the intermediate picture the metric on the brane is fluctuating i.e. we are integrating over it in the path integral. But it is not clear if this metric has its own kinetic term (for example the Einstein-Hilbert term) or not. Interestingly, it was proposed in [53] that replacing Dirichlet boundary condition to Neumann boundary condition in the AdS/CFT promotes the boundary metric to a dynamical field in the path integral description but doesn't add any kinetic term of it. Hence, in our case there is no kinetic term for the metric in the intermediate picture. This can be seen by the following consideration. We can understand the perturbative Hilbert space (the Fock space) either from the intermediate picture point of view or the bulk point of view. They are equivalent since the intermediate picture and the bulk picture are dual to each other. To construct this Hilbert space, we can do the perturbative canonical quantization in the intermediate picture. When we consider the situation where the metric is dynamical, if there is no kinetic term for the metric then we don't have creation and annihilation operators for the graviton. Hence in this case we have equal number of creation and annihilation operators to construct the Fock space representation of the perturbative Hilbert space as in the case where the metric is not dynamical. Nevertheless, if we have a kinetic term for the metric then we have creation and annihilation operators for the graviton when the metric is dynamical. Therefore, the (perturbative) Hilbert space would be larger than the case where the metric is not dynamical. Meanwhile we can also construct the Hilbert space by doing canonical quantization in the bulk picture. For the bulk picture described by the action Equ. (1.1) with Neumann boundary condition Equ. (2.2) for the bulk metric fluctuation, if we do canonical quantization then the number of creation and annihilation operators associated with the bulk metric is the same in the Dirichlet boundary case. The reason is that the Neumann boundary condition in this case is equally as restrictive as the Dirichlet boundary condition. This is because in the Neumann case the second equation in Equ. (2.3) restricts the bulk metric fluctuation $\delta g_{\mu\nu}$ to satisfy $\partial_n \delta g_{\mu\nu} - \frac{2T}{d-1}\delta g_{\mu\nu} = 0$ near the brane and in the Dirichlet case $\delta g_{\mu\nu}$ is restricted to be zero near the brane so the two boundary conditions are just projecting into different sets of bulk modes with equal dimensionality. We call these two sets of modes $MD$ and $MN$. Furthermore, if we have a DGP term on the brane in the bulk picture then the Hilbert space for the Dirichlet boundary condition case is the same as the case without the DGP term as the bulk modes satisfying the bulk equation of motion and the boundary condition are still $MD$. Nevertheless, if we consider now the Neumann boundary condition case then the boundary equation of motion for the bulk metric fluctuation $\delta g_{\mu\nu}$ is naively given by the linearized equation

$$\partial_n \delta g_{\mu\nu} - \frac{2T}{d-1}\delta g_{\mu\nu} - \frac{G_{d+1}}{G_d}(\text{d-dimensional linearized Einstein's equation for } \delta g_{\mu\nu}) = 0\,.$$
(4.1)

So a bulk metric fluctuation $\delta g_{\mu\nu}$ can be expanded using the modes from $MD \cup MN$. Let's

schematically denote such a fluctuation by

$$\hat{\delta g}_{\mu\nu} = \hat{A}_D \delta g_{\mu\nu}^D + \hat{A}_N \delta g_{\mu\nu}^N \,, \tag{4.2}$$

where $\hat{A}_{D(N)}$ denotes the creation and annihilation operators in the Dirichlet (Neumann) sector. This is the standard expression in the canonical quantization. Nevertheless, we should be careful with the boundary equation of motion Equ. (4.1) as obviously Equ. (4.2) doesn't satisfy Equ. (4.1). More precisely the left hand side of Equ. (4.1) reads (near the brane)

$$\hat{A}_D \partial_n \delta g_{\mu\nu}^D - \hat{A}_N \frac{G_{d+1}}{G_d} (\text{d-dimensional linearized Einstein's equation for } \delta g_{\mu\nu}^N) \,. \tag{4.3}$$

There are two obstructions for this to be zero. The first is that $\hat{A}_D$ and $\hat{A}_N$ are supposed to be independent operators. The second is that in the one-brane case the bulk analysis implies that $\delta g_{\mu\nu}^N$ are massive graviton modes [1] therefore

$$(\text{d-dimensional linearized Einstein's equation for } \delta g_{\mu\nu}^N) = m^2 \delta g_{\mu\nu}^N \,, \tag{4.4}$$

where $m^2$ is the mass for the given mode $\delta g_{\mu\nu}^N$ (different modes $\delta g_{\mu\nu}^N$'s would have different mass). The first obstruction is easy to understand as the equation of motion Equ. (4.1) is actually an interacting equation (the interaction strength is controlled by $\frac{G_d}{G_{d+1}}$) and for the (perturbative) canonical quantization (in the weak interaction regime) we can forget about the interaction when we are constructing the Fock space. Therefore, we can isolate the second order derivative terms in Equ. (4.3) which is exactly the second term. Naively the vanishing of this term is the interaction free equation of motion that is used construct the Fock space. Nevertheless, the second obstruction tells us that this expectation cannot be satisfied and the correct free equation of motion we can use is a massive equation. Hence, in the one-brane case the DGP term should be introduced together with a graviton mass term. This ensures Equ. (4.4) which is a consistency condition. However, the choice of the mass parameter is not arbitrary and it is controlled by the bulk analysis. For the consistency with the picture that the graviton localized on the Karch-Randall brane is the first massive mode $\delta g_{\mu\nu}^{N(1)}$ in $MN$ [1], we have to choose $m^2$ to be $m_1^2$ where $m_n^2$ $(n = 1, 2, 3 \cdots)$ is the mass square for the tower of modes in $MN$. This tells us that the mode functions that we can use to construct the Fock space are those from $MD \cup \delta g_{\mu\nu}^{N(1)}$. For each of them we have the associated creation and annihilation operators. Hence now we see that the Fock space (the perturbative Hilbert space) is larger than the Dirichlet boundary condition case. This is also consistent with the result in the intermediate picture (i.e. the modes are the original field theory (no metric fluctuation) modes with the massive graviton modes (for a specific mass)).

From the above analysis, we see that the precise description of the intermediate picture of the original version of the Karch-Randall braneworld Equ. (1.1) is given by the following path integral for the brane part[7]

$$Z_{\text{intermediate}}^{\text{brane}} = \int [Dh_{\mu\nu}][D\phi_{\text{matter}}]_h e^{iS_{\text{CFT}+T\bar{T}\text{-like deformation}}[\phi; h]} \,. \tag{4.5}$$

---

[7] For some evidence of this result see [50, 51].

Here we emphasize that the matter field measure also depends on the metric and this is important for subtle quantum effect such as graviton mass [54]. Again the full intermediate picture includes also a bath and that can be easily included with Equ. (4.5) equipped. The background metric is a solution of the induced gravitational equation

$$\langle T_{\mu\nu}^{\text{matter}} \rangle = 0 \,, \tag{4.6}$$

which is the saddle point approximation of the metric integration in Equ. (4.5). Moreover, the DGP term on the brane can be now understood as adding a kinetic term for the metric $h_{\mu\nu}$ in the intermediate picture (i.e. add $\frac{1}{16\pi G_d} \int d^d x \sqrt{-g} R[h]$ to the exponent in Equ. (4.5)) meanwhile a proper graviton mass term should also be added for the consistency with analysis in [1]. Without the DGP term the replica entropy on the brane is the matter field entanglement entropy (for a subregion which can be defined diffeomorphism invariantly for example the exterior of the black hole, see Sec.4.3).

## 4.2   The Derivation of the Modified Island Formula

In this section, we discuss the derivation of the formula Equ. (2.6) as well as Equ. (2.8). For simplicity, we only consider the case without the DGP terms i.e. Equ. (2.6). Then the result Equ. (2.8) in the case with the DGP terms is easily obtained resorting to the calculation in [55].

Firstly, let's consider the case of a fixed subregion $\mathcal{R}$. The entanglement entropy of $\mathcal{R}$ can be calculated using the replica trick

$$S_{\mathcal{R}} = \lim_{n \to 1} \frac{1}{1-n} \ln \text{tr} \left( \rho_{\mathcal{R}}^n \right) = \lim_{n \to 1} \frac{1}{1-n} \ln Z_n \,, \tag{4.7}$$

where $\rho_{\mathcal{R}}$ is the reduced density matrix of the subregion $\mathcal{R}$. The appearance of the entanglement island is due to the the replica wormhole contribution to the gravitational path integral [56] representation of the replicated partition function $Z_n$. It is easy to see that we would get $S_{\text{matter}}(\mathcal{R} \cup \mathcal{I})$ with island included. The question in our case is how to understand the $\min_{\mathcal{I}} \text{ext}$. In the case that we have kinetic terms for the metric it is the result of the saddle point approximation of the gravitational path integral [46, 55, 57] where $\min_{\mathcal{I}} \text{ext}$ comes from solving the resulting backreacted gravitational equation (and the entropy functional will be $S_{\text{matter}}(\mathcal{R} \cup \mathcal{I})$ together with additional geometric terms due to the nontrivial kinetic terms for the metric in the total action). In our case, this can be understood by the interesting property that the replica entropy is RG invariant [45]. In other words, to compute $S_{\mathcal{R}}$ (with $\mathcal{R}$ fixed) we can firstly integrating out the matter fields in the intermediate picture which would give us an effective gravitational action $S_{eff}[h]$ [58].[8] The resulting effective action is a higher curvature gravity where the higher curvature expansion is controlled by the cutoff scale of the matter fields. The leading order term is a cosmological constant and the next the leading order term is the Einstein-Hilbert term [59]. The RG invariance of the replica

---

[8]For free fields $S_{eff}[h]$ can be computed by the heat kernel method [59].

entropy tells us that $S_{\text{matter}}(\mathcal{R} \cup \mathcal{I})$ equals to the generalized Wald entropy $S_{grav}(\mathcal{R} \cup \mathcal{I})$ of the region $R \cup I$ [57] computed by $S_{eff}[h]$ as both of them are equal to the replica entropy of the subregion $\mathcal{R} \cup \mathcal{I}$. Moreover, as a gravitational theory the resulting $S_{\mathcal{R}}$ computed by $S_{eff}[h]$ is given by

$$S_{\mathcal{R}} = \min_{\mathcal{I}} \text{ext} S_{grav}(\mathcal{R} \cup \mathcal{I}) \,, \tag{4.8}$$

for the same reasons as in AdS/CFT [46, 55, 57]. So using the RG invariance of the replica entropy we have the result

$$S_{\mathcal{R}} = \min_{\mathcal{I}} \text{ext} S_{\text{matter}}(\mathcal{R} \cup \mathcal{I}) \,. \tag{4.9}$$

Here we emphasize that the right hand side is computed in the background geometry as described in Fig.3 and it is a result of the saddle point approximation of the path integral over metric as discussed at the end of Sec.4.1.

Nevertheless, we've ignored the fact that $\mathcal{R}$ is a subregion in a gravitational universe. The definition of $\mathcal{R}$ should be consistent with diffeomorphism invariance and holography. A natural choice would be that before we perform the path integral over the metric in the universe (i.e. the gravitational bath) that contains $\mathcal{R}$ we define $\mathcal{R}$ for each geometry by taking $\min_{\mathcal{R}} \text{ext}$ of $S_{\text{matter}}(\mathcal{R} \cup \mathcal{I})$ in that geometry. As a result, a diffeomorphism invariant quantity we get is

$$S = \min_{\mathcal{I}, \mathcal{R}} \text{ext} S_{\text{matter}}(\mathcal{R} \cup \mathcal{I}) \,, \tag{4.10}$$

and it is the entanglement entropy of the resulting radiation region $\mathcal{R}_{\min}$ from the extremization and minimization. The right hand side in this formula is computed in the geometry of Fig.3 as a result of the saddle approximation of the metric integration and $\mathcal{R}_{\min}$ lives on this geometry as well.

As we discussed in Sec.2.2 this formula tells us that both the entanglement island $\mathcal{I}$ and the radiation region $\mathcal{R}$ are empty. This is a direct consequence of diffeomorphism invariance.

## 4.3 Robustness of the Result Under Coarse Graining

There are possible coarse-graining protocols that we could have entanglement island in the intermediate picture of the wedge holography model.

There are two types of such protocols [23, 24]. The first type of protocols is to turn off gravity on the universe that contains the radiation region with the hope that we wouldn't have to impose diffeomorphism constraints and so able to specify arbitrary $\mathcal{R}$ [23]. Then the calculation reduces to the standard calculation of a nongravitational bath [12] where the entanglement island and nontrivial Page curve emerge. The second type of protocols is to specify $\mathcal{R}$ in a diffeomorphism invariant way which however could give a nonvanishing result for $\mathcal{R}$. An example is [24] where the boundary $\partial \mathcal{R}$ of $\mathcal{R}$ is defined by firstly choosing a time band $\Gamma$ on the asymptotic boundary and then taking the domain of dependence of the complement of $\mathcal{R}$ as the bulk (the bulk of the gravitational bath) domain of dependence of this time band $\Gamma$. This is done for each metric on the bath that we are integrating over (see Fig.6). Then using the saddle point approximation for the metric integration we get $\mathcal{R}$ as specified by the above protocol on the black hole geometry (see Fig.7).

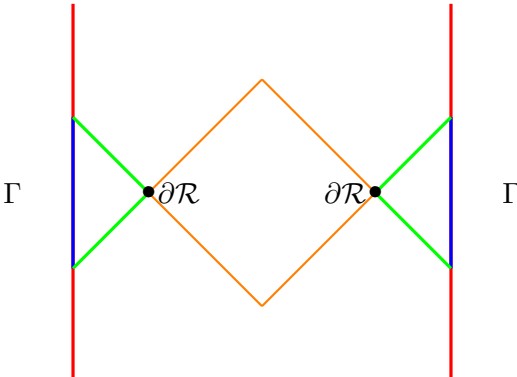

**Figure 6**: A demonstration of the protocol proposed in [24] to specify $\mathcal{R}$ in a gravitational universe diffeomorphism invariantly. The red boundary lines are the fixed asymptotic boundaries where $\Gamma$ (the blue intervals) is a time band and the bulk geometry is not fixed. The proposal is that for each bulk geometry we specify $\mathcal{R}$ in the following way. We first determine the bulk domain of dependence of $\Gamma$ (the bulk region enclosed by $\Gamma$ and the green lines) whose boundary is $\partial\mathcal{R}$ and the domain of dependence of $\mathcal{R}$ is determined as shown in the figure as the interior of the orange diamond. Then we have to integrate over the geometries.

However, there are various problems with these coarse-graining protocols. The first class requires a consistent decoupling of gravity in a gravitational theory. In Einstein's gravity this is usually claimed to be achieved by sending the Newton's constant $G_N$ to zero which will localize the path integral over the metric to the metric that satisfies Einstein's field equations and with fluctuations suppressed.[9] In our case, the gravity is induced and the induced Newton's constant is controlled by the cutoff scale of the matter field. Hence to decouple gravity in this way we have to tune up the cutoff scale of the matter field. Hence, in the bulk description, the consistent way to decouple gravity is to dial the bath brane all the way to the conformal boundary. And we know that when the gravity is decoupled in the bath in this way the graviton in the gravitational universe (the brane) that is coupled to the bath becomes massive [1, 10]. The second class relies on the specification of a boundary time band $\Gamma$ which however wouldn't be consistent with holography. The reason is that due to holography the boundary has a unitary dynamics with a time evolution generator and so the time band essentially encode all the physics of the boundary [61–65]. Or in other words a consistent choice of the time band $\Gamma$ with holography would be to choose $\Gamma$ as the whole boundary. This will again give $\mathcal{R} = \emptyset$.[10]

---

[9]Up to the subtlety of the introduction of free graviton [60].

[10]Including the Hamiltonian into the algebra is important for holography otherwise many techniques in quantum gravity which makes use of the existence of a time evolution, such as the gravitational path integral, would breakdown.

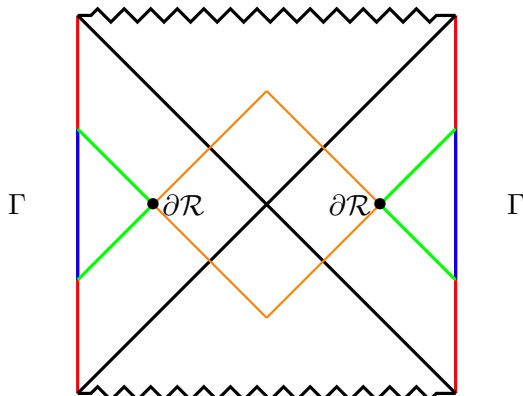

**Figure 7**: For simplicity, we only show the Penrose diagram of the gravitational bath. The saddle point approximation of the metric integral Equ. (4.6) determines the background geometry to be a black hole. The subregion $\mathcal{R}$ in this diagram is determined by the protocol as discussed in Fig.6.

## 5 Conclusion

In this paper, we provide a careful check of the conclusion that entanglement island is not consistent with massless gravity in a model of massless gravity constructed using the Karch-Randall braneworld- the wedge holography model. This model is doubly holographic and it has three equivalent descriptions. We discovered that the consistency of holography in this model provides interesting bounds on the DGP parameters. Moreover, we provide a careful analysis of the intermediate description of this doubly holographic model and we see that it is described by an induced gravity theory with the matter field as a $T\bar{T}$-like deformation of the $\mathrm{CFT}_d$ that duals to the gravity in the $\mathrm{AdS}_{d+1}$ bulk (with no brane). We also show that to match the bulk analysis a graviton mass term should be introduced when we introduce a DGP term and the mass square is given by that of the first normalizable KK mode for bulk graviton with Neumann boundary near the brane. We used the intermediate picture to show that entanglement island doesn't exist due to diffeomorphism invariance. At the end, we discussed two classes of coarse-graining protocols that could relax the constraint from diffeomorphism invariance on the existence of entanglement island. We found that defining them consistently is subtle and once they are defined consistently we either lose entanglement island or we modify the gravitaitional theory on the gravitational universe (the brane)that is coupled to the bath to be massive .

### Acknowledgements

We thank Andreas Karch, Carlos Perez-Pardavila, Suvrat Raju, Lisa Randall, Marcos Riojas, Sanjit Shashi and Merna Youssef for relevant collaborations. We thank Amr Ahmadain, Juan Maldacena, Rongxin Miao, Rashimish K. Mishra, Dominik Neuenfeld and Yasunori Nomura

for relevant discussions. This work is supported by the grant (272268) from the Moore Foundation "Fundamental Physics from Astronomy and Cosmology".

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
