# Peer review of "Entropy, Entanglement Islands and Swampland Bounds in the Karch-Randall Braneworld"

_SciPost Physics_

## Round 1 · Referee Report · Anonymous (Referee 1) · 2024-1-26

Strengths

1- The questions addressed in this paper are very interesting and relevant. 2- The motivation and the result the author would like to derive are clearly stated.

Weaknesses

1- Neither concrete computation nor rigorously provable statement is presented. 2- The main result is based on qualitative arguments, where many of them are cited from other literature. However, some crucial arguments cannot be justified by the cited references. Therefore, the validity is not well-established. Details will be presented in the main body of the report.

Report

This paper is in the stream of a sequence of papers written by the current author (including Ref. [5, 11, 13, 28]), where it is argued that the existence of an entanglement island may be inconsistent with the existence of massless graviton from different perspective. As a whole, the excellent sequence of papers Ref. [5, 11, 13, 28] is interesting and highly valid, and the questions asked are important and fundamental. Therefore, the questions addressed in this paper are interesting and highly relevant by its very nature.

Throughout the paper, the author argued that Eq. (2.6) and Eq. (2.8) should be the proper modification of the island formula that one should consider and applying them to "physical" setups would necessarily lead to the conclusion that there is no island at all. After explaining the basic setup in section 2, the author explained what "physical" means here in section 3, provided some qualitative arguments in section 4.1 & 4.3, and presented a "derivation" of Eq. (2.6) and Eq. (2.8) in section 4.2.

However, compared to the other papers in this sequence, the contributions made in this paper is unclear. The essential parts of the argument are, to my knowledge, already discussed in the author's excellent paper Ref. [11]. The idea that one may use the positivity of the entanglement entropy and the causal wedge inclusion to constrain the DGP term localized on the brane presented in section 3 is interesting. However, no concrete computation or analysis is presented here. (Although there are computations in the author's excellent paper Ref. [28].) The "derivation" presented in section 4.2 is new. Therefore, I will consider this part as the main result of the current paper in the following.

The "derivation" presented in section 4.2 is new and would be of crucial relevance if it was valid. However, the arguments are just qualitative and does not seem to be valid. Below, I will list some, but not all, arguments whose validity is skeptical, followed by some minor issues I found when reading the paper.

1- At the end of page 12, “Hence, the matter field in the intermediate picture is the holographic dual of the cutoff AdSd+1 bulk and it is described by a T ¯ T -like deformation of the CFTd on the brane”. The author’s argument is insufficient to support this statement. In fact, the TTbar deformation and the brane are distinct from each other, in the sense that the boundary of the TTbar deformed theory is Dirichlet [47], while in the brane case it is Neumann.

2- In footnote 6, the author argued that their derivation of the entropy formula is supported by the Lewkowycz-Maldacena's argument. However, the Lewkowycz-Maldacena argument cannot be applied here. This is again because the boundary condition on the brane is Neumann, while the Lewkowycz-Maldacena's argument is valid only when the boundary condition is Dirichlet.

3- In the derivation of eq. (4.7), the author implicitly used the Lewkowycz-Maldacena argument on the $R$ side at the first step, which requires the Dirichlet boundary condition.

Below, please let me list some minor issues I found when reading the paper:

1- Eq. (2.2) does not necessarily imply the boundary condition is the Neumann type. Mixed boundary conditions also have the same feature. It seems that $n_i$ is missed somewhere in Eq. (2.2).

2-Below Eq. (2.5), the author said, “The two branes will intersect each other at u = 0 (a (d-1)-dimensional manifold) on the asymptotic boundary.” 
However, the two branes are parallel at the asymptotic infinity and never intersect each other.

3- The author said that the motivation for considering Eq. (2.6) is because with Eq. (2.6), $\mathcal{R}$ is “well-defined and consistent with diffeomorphism invariance”. However, the reasons are not explained. Also, is (2.6) is the only possibility consistent with diffeomorphism, or is it just one specific choice which is consistent with diffeomorphism?

4- Below fig.3, the author said “Equ. (2.7) as it computes the fine-grained entanglement entropy of the subregion R”. How can there be a fine-grained entropy in the intermediate picture while the intermediate picture itself is just a low energy effective one?

5- In the last paragraph on page 9, “be smaller that …” is considered to be “be smaller than …"

6- In the 2nd line of page 11, the author said, “This condition exclusively determines that γmin = H.”. However, it is possible that $\gamma_{min}$ lies in the causal shadow of the defect.

In summary, the current paper lacks valid contributions and I do not think it is qualified to be published on scipost. However, the idea addressed in this paper is interesting and the questions are important. It would be great if the author can present some solid analyses if they consider a resubmission of this paper. One concrete thing the author might be able to do is to present a Lewkowycz-Maldacena type of analysis to the TTbar deformed AdS/CFT, which is still different from the brane case the author would like to finally address, but analytically more contractable.

  • validity: -
  • significance: -
  • originality: -
  • clarity: -
  • formatting: -
  • grammar: -

Author:  Hao Geng  on 2024-01-27  [id 4288]

(in reply to Report 1 on 2024-01-26)

I thank the referee for the detailed report.

Before going to details, I'd like mention that the goals of this paper are to review the recent progress in our understanding of quantum gravity using braneworld, articulate important subtleties in the braneworld models including the precise description of its intermediate picture and outlining interesting questions for future studies. The paper [28] is a follow up paper of this one, studying some of the interesting questions proposed to be studied in this paper, with many junior students under our supervision.

Here are my reply to the referee's main comments 1-3 with some detailed explanations for relevant points which hopefully could address the referee's confusions on the literature.

  1. It's true that in standard AdS/CFT correspondence the TTbar-like deformations for CFTs works for Dirichlet boundary conditions of the bulk metric fluctuations near the cutoff surface (or brane). The essentially reason for this statement is that the question under that study is a field theory question in other words the field theory dual is not gravitational. Hence the metric fluctuation is just zero in that field theory system. This says that the bulk metric fluctuation near the cutoff surface should also be zero as they are holographically dual. The take-home message from the AdS/CFT study of the TTbar deformation is that there is a precisely way to move the AdS boundary "into" the bulk in the CFT description and the dual of the depth of this move into the bulk is the TTbar deformation parameter. Nevertheless, this doesn't obstruct us from studying the AdS dual of the operation that turns on gravity on the dual field theory. Turning on gravity or coupling the TT-bar deformed CFT to gravity just means adding another dynamical modes- the metric fluctuation in the original field theory system. Then the question is what is the AdS dual of this new system? The answer is articulated in Section 4.1 and it is simply that we just have to add the modes to the bulk metric that satisfies the Neumann boundary condition and at the same time keep those Dirichlet modes. We notice that these two sets of modes are not contradicting each other and they are independently defined. Each of them dual to different sectors in the TTbar deformed CFT+gravity system. The Dirichlet modes dual to the TTbar deformed CFT and the Neumann modes dual to the dynamical metric (see Equ. (4.2) for example).

  2. I think this comment is relevant to 1. It is indeed true that Lewkowycz-Maldacena works for Dirichlet boundary condition and the reason is that it is computing the dual matter field entanglement entropy. Our study didn't contradict this point. As I explained in 1 that when we couple the TTbar deformed CFT to gravity, the TTbar deformed CFT still duals to bulk Dirichlet modes and the entanglement entropy of this TT bar deformed CFT sector can still be computed or derived holographically using the Lewkowycz-Maldacena procedure. The real question is what happens to the gravitational part? In Einstein's gravity we know that there should an area term A/4G but if I use RT formula I'll see that the RT surface area for a chosen bipartition doesn't care about boundary conditions. This observation motivates the proposal that the gravity on the brane is an emergent gravity theory. The meaning of this is explained in Section 4.1 that in the path integral of the TTbar deformed CFT system we also integrate over metric but the metric doesn't have a kinetic term like Einstein-Hilbert term on the brane. Hence the formula for the subregion entropy stays the same in the case with no gravity which is exactly the situation in the RT calculation. Moreover, this is also consistent with the usual understanding that the Einstein-Hilbert term on the brane should be introduced by hands by adding the DGP term.

  3. I think this is the same as comment 2.

Here are my reply to the referee's minor comments 1-6.

  1. I intended to call delta g_munu \neq0 Neumann-type boundary condition. Maybe this is a confusing and unconventional nomenclature so I should articulate this carefully.

  2. I thank referee to point this out. I agree with the referee that if think of the defect should be regulated by an interval then they are in fact parallel. Though I followed our language in [5, 11, 13, 28] to consider unregulated defect or exact wedge holography.

  3. I thank the referee to bring up this interesting point. This question regards how to define a subregion entropy in a gravitational universe. I addressed this question in section 4.3 where I showed that we should take the logic as that we need to start with a diffeo-invariant formulation of the question for the subregion entropy in a gravitational universe and study it in braneworld. In this way, (2.6) is a well defined question to start with and what we did is to study this question using holography. This question is also recently analyzed in https://inspirehep.net/literature/2737271 whose answer is indeed our proposal.

  4. I think low energy EFT doesn't contradict with fine-grained entropy. This point is articulated in section 4.1. The interesting thing is that we have to consider the EFT in a curved spacetime background. In this way, we can see that the entanglement entropy of a simple bipartition is RG invariant. The reason is that integrating out high energy modes generates Einstein-Hilbert terms which contributes to EE in the replica-trick calculation. See https://arxiv.org/abs/hep-th/9506182 for example. This is an old idea from Susskind and Uglum saying the matter EE renormalized Newton's constant.

  5. I thank the referee pointing this out. I should use "than" than "that".

  6. I thank the referee for this interesting question. I believe at least in the geometry discussed in the paper gamma_min=H is the only solution. There might be other interesting solutions in other geometries.

In the reply above, I believe I carefully addressed the referee's confusions and questions.

I believe the referee's complain that "The main result is based on qualitative arguments, where many of them are cited from other literature. However, some crucial arguments cannot be justified by the cited references. Therefore, the validity is not well-established. Details will be presented in the main body of the report." is based on the confusion of boundary conditions. This is also a common confusion even for people working in the field which spent hard time to articulate in this paper. I hope this part has been resolved.

I believe the referee's complain that "Neither concrete computation nor rigorously provable statement is presented." is partially based on the previous confusion. Again I should emphasize that the goal of this paper is not to carry out any new computations but to resolve several subtleties in the braneworld model which are even confusing for people working in the field. Hence I tried not to do the demonstration by complicated computation and instead use simple examples as long as I can. Meanwhile I defer the study of interesting questions proposed in this paper to future studies and an example is [28].

---

## Round 1 · Referee Report · Anonymous (Referee 2) · 2024-2-29

Strengths

  1. The paper tackles difficult yet very interesting problem of defining entanglement in gravitating systems.
  2. It is an interesting direction to study relation between swampland and the island formula.

Weaknesses

  1. The formula for entanglement entropy in gravitating system this the paper is based on lacks both physical and information theoretic interpretations.
  2. The swampland bound proposed in this paper does not seem to be strong.

Report

This paper studies an extension of the island formula for gravitating subregions by considering the Karch-Randall braneworld models. In the author's previous paper, such an extension was proposed as a usual island formula with varying subregions, minimizing the whole quantity. As a consequence, the island and the dynamical subregion are both empty. Thus, the defined quantity is zero. The author also considers adding the Einstein-Hilbert action on the Karch-Randall branes with tunable Newton constants (DGP term parameters). The author found that, again, an empty island and a dynamic subregion. Furthermore, the author used this result to constrain the DGP term parameters, which the author calls the swampland bounds.

There are several puzzling features that need to be clarified and addressed before considering publication in SciPost. I will explain them in the following.

The first and most important question is, what is the physical and information-theoretic meaning of the proposed formula eq (2.6) and (2.8)? The most puzzling aspect of these formulas is what the quantum state is referred to in these formulae. If the subregion R is moving freely, it is hard to identify what the degrees of freedom are being entangled. This is not the case in the usual island formula, which talks about matter state on a non-gravitating system. Since the quantum state is undefined, it is hard to associate information theoretic meaning. At the moment, I can only understand these formulae as a diff-invariant, mathematical generalization of the island formula.

The second point is a question on the setup. It seems the setup requires wedge holography with two AdS boundaries (interpreting two red dots in Fig 4 as two boundaries). Such setup was hard to construct in AdS/BCFT due to brane intersections. It would be nice to have an explanation why there is such a difference.

The third point is the question on the significance of the conditions in section 3. How The terms on the left-hand side seem all non-negative.

---

## Editorial Decision

resubmitted